# Inhibitory Mechanisms of Lusianthridin on Human Platelet Aggregation

**DOI:** 10.3390/ijms22136846

**Published:** 2021-06-25

**Authors:** Hla Nu Swe, Boonchoo Sritularak, Ponlapat Rojnuckarin, Rataya Luechapudiporn

**Affiliations:** 1Pharmaceutical Sciences and Technology Program, Faculty of Pharmaceutical Sciences, Chulalongkorn University, Bangkok 10330, Thailand; hlanu.ph@gmail.com; 2Department of Pharmacognosy and Pharmaceutical Botany, Faculty of Pharmaceutical Sciences, Chulalongkorn University, Bangkok 10330, Thailand; Boonchoo.Sr@chula.ac.th; 3Natural Products for Ageing and Chronic Diseases Research Unit, Chulalongkorn University, Bangkok 10330, Thailand; 4Department of Medicine, Faculty of Medicine, Chulalongkorn University, Bangkok 10330, Thailand; rojnuckarinp@gmail.com; 5Department of Pharmacology and Physiology, Faculty of Pharmaceutical Sciences, Chulalongkorn University, Bangkok 10330, Thailand

**Keywords:** lusianthridin, antiplatelet aggregation, ADP, arachidonic acid, cyclooxygenase enzymes, cAMP

## Abstract

Lusianthridin is a phenanthrene derivative isolated from *Dendrobium venustum*. Some phenanthrene compounds have antiplatelet aggregation activities via undefined pathways. This study aims to determine the inhibitory effects and potential mechanisms of lusianthridin on platelet aggregation. The results indicated that lusianthridin inhibited arachidonic acid, collagen, and adenosine diphosphate (ADP)-stimulated platelet aggregation (IC_50_ of 0.02 ± 0.001 mM, 0.14 ± 0.018 mM, and 0.22 ± 0.046 mM, respectively). Lusianthridin also increased the delaying time of arachidonic acid-stimulated and the lag time of collagen-stimulated and showed a more selective effect on the secondary wave of ADP-stimulated aggregations. Molecular docking studies revealed that lusianthridin bound to the entrance site of the cyclooxygenase-1 (COX-1) enzyme and probably the active region of the cyclooxygenase-2 (COX-2) enzyme. In addition, lusianthridin showed inhibitory effects on both COX-1 and COX-2 enzymatic activities (IC_50_ value of 10.81 ± 1.12 µM and 0.17 ± 1.62 µM, respectively). Furthermore, lusianthridin significantly inhibited ADP-induced suppression of cAMP formation in platelets at 0.4 mM concentration (*p* < 0.05). These findings suggested that possible mechanisms of lusianthridin on the antiplatelet effects might act via arachidonic acid-thromboxane and adenylate cyclase pathways.

## 1. Introduction

Cardiovascular diseases (CVD) are the leading global cause of morbidity and mortality. According to WHO, more than 15.2 million people died from CVD in 2016 [1]. The development of CVD is associated with the formation and rupture of atherothrombotic plaques, which are mediated by many factors including lipid peroxidation, foam cell formation, dysfunction of vascular endothelial cell, platelet activation, and depletion or deposition of chemical mediators [2].

Platelet activation is an essential factor for thrombus formation. Platelets maintain hemostasis by preventing blood loss and supporting vascular integrity. When the endothelial layer from a blood vessel is broken, fibers are exposed after an injury or plaque rupture. Collagen attracts platelets like a magnet and induces platelet adhesion, activation, and aggregation to provide hemostatic plugs or thrombus formation [3]. Additionally, hypersensitive platelets may lead to CVD, such as thrombosis, atherosclerosis, ischemic stroke, and myocardial infarction. 

Therefore, antiplatelet agents are used for preventing and treating CVD to reduce mortality [2]. However, antiplatelet therapies still have some limitations. Aspirin, a commonly used standard antiplatelet agent, has side effects such as gastric ulcers, bleeding, and renal failure. Another more potent agent, clopidogrel, may cause bleeding with high cost [4]. Therefore, searching for newer antiplatelet agents from medicinal plants is an attractive field. Orchids are used as traditional Chinese medicine and pharmacological activities of their chemical constituents have been studied recently [5,6,7,8,9,10]. The most represented genera of orchids were *Dendrobium* spp. (183 species) [11,12]. The methanol extract of *Dendrobium loddigesii* was reported to inhibit washed rabbit platelets induced by arachidonic acid [6]. Two phenanthrenes isolated from the stems of *Dendrobium longicornu* exhibited weak antiplatelet aggregation activities in platelet-rich plasma (PRP) from New Zealand white rabbits in vitro [13]. Chen et al. 2000 also showed that the phenanthrene derivatives; 3,7-dihydroxy-2,4-dimethoxyphenanthrene and erianthridin extracted from *Ephemerantha lonchophylla* exhibited significant inhibitory activities against washed rabbit platelets aggregation stimulated by arachidonic acid with estimated the half-maximal inhibitory concentration (IC_50_) values of 24 and 9 µM, respectively [14]. In this study, we were interested in lusianthridin, a phenanthrene derivative isolated from *Dendrobium venustum*. The turbidimetric method is widely used for investigations of antiplatelet aggregation in human platelet-rich plasma. Since clopidogrel needs to be biotransformed into active metabolites in vivo and native clopidogrel does not inhibit ADP-induced platelet aggregation in platelet-rich plasma [15], aspirin was used as positive control in the present study. The study focused on the inhibitory effect of lusianthridin on human platelet aggregation induced by various agonists and investigated the possible modes of actions of lusianthridin on platelets. 

## 2. Results

### 2.1. Antiplatelet Aggregation Activities of Lusianthridin 

As shown in Figure 1A, lusianthridin more selectively inhibited the second phase of ADP-induced platelet aggregation compared with the first phase. For arachidonic acid-induced platelet aggregation, lusianthridin at 0.0125 mM significantly increased the delaying time compared with the control group (80.00 ± 8.91 vs. 46.25 ± 5.59 s, respectively, *p* < 0.05). However, the aggregating inhibition was not observed at this concentration (Figure 1B). As shown in Figure 1C, we found that lusianthridin at 0.4 mM concentration could significantly prolong the lag time of collagen-induced platelet aggregation compared with vehicle control (114.45 ± 15.82 vs. 68.32 ± 5.00 s, respectively, *p* < 0.05). Lusianthridin inhibited platelet aggregation induced by arachidonic acid, collagen, and ADP with different sensitivities as determined by the IC_50_ of 0.02 ± 0.001 mM, 0.14 ± 0.018 mM, and 0.22 ± 0.046 mM, respectively (Figure 2).

### 2.2. Molecular Docking Studies of Lusianthridin on COX-1 and COX-2 Enzymes

As shown in the Figure 3A, the binding site of lusianthridin was partially similar to the binding site of arachidonic acid on the cyclooxygenase-1 (COX-1) enzyme. Arachidonic acid bound the active site of COX-1 with two hydrogen bonds (Arg-120 and Phe-470), one carbon-hydrogen bond (Gly-471), and three hydrophobic bonds (Val-116, Leu-531, and Ala-527) (Figure 3B). Moreover, its binding site existed at the vicinity of amino acid residues Glu-524, Ile-89, Leu-93, and Tyr-355. Lusianthridin bound the COX-1 enzyme by a pi-donor hydrogen bond with Tyr-355, two hydrophobic bonds with Val-116, and Ile-89 and a hydrophobic bond with Leu-93 (Figure 3C). Arg-120 and Glu-524 surrounded the binding site of lusianthridin. Additionally, the binding affinity of lusianthridin was comparable to that of arachidonic acid (−7.2 and −7.9 kcal/mol, respectively).

Figure 4A represents the interaction of lusianthridin and arachidonic acid with the cyclooxygenase-2 (COX-2) enzyme. We found that the binding site of lusianthridin was far from that of arachidonic acid. Arachidonic acid bound with the COX-2 enzyme via the key amino acid residue, Arg-120 (Figure 4B). When lusianthridin was docked at the same site, it formed two hydrogen bonds with Arg-44 and Asn-39 and three hydrogen bonds with Pro-153, Leu-152, and Cys-47 (Figure 4C). In addition, its binding site existed near the amino acid residues Glu-465 and Arg-469. Lusianthridin showed a higher binding affinity than arachidonic acid (−9.3 vs. −7.3 kcal/mol, respectively).

### 2.3. Effects of Lusianthridin on Cyclooxygenase Enzymes Activities

The effect of lusianthridin on cyclooxygenase activity was determined using COX fluorescent inhibitor screening assay kit. Lusianthridin showed concentration-dependent inhibitory effects on both COX-1 and COX-2 enzymatic activities (Figure 5A,B) with the IC_50_ value of 10.81 ± 1.12 and 0.17 ± 1.62 µM, respectively (Appendix A). 

### 2.4. Effect of Lusianthridin on cAMP Levels

To assess the cyclic adenosine monophosphate (cAMP) level in platelets, the experiment was carried out by using the cAMP ELISA kit. As shown in Figure 6, ADP 4 µM significantly decreased the cAMP level in platelets (7.04 ± 0.21 pmol/108 platelets) compared to the basal cAMP level (11.01 ± 0.43 pmol/108 platelets). When platelets were pre-incubated with lusianthridin at the concentration of 0.4 mM, the decrease of ADP-induced cAMP level in platelets was prevented (12.21 ± 1.42 pmol/108 platelets, *p* < 0.05). The 3-isobutyl-1-methylxanthine (IBMX) was used as a positive control and the results of lusianthridin were very similar to those of IBMX (11.66 ± 1.98 pmol/108 platelets).

## 3. Discussion

In the present study, we evaluated the possible mode of actions of lusianthridin in human platelets. Lusianthridin inhibited platelet aggregation stimulated by arachidonic acid, collagen, and ADP with different potencies. Additionally, molecular docking and COX activity assay revealed that lusianthridin inhibited the COX-1 enzyme and the COX-2 enzyme. Moreover, lusianthridin prevented the decrease of ADP-induced cAMP level in platelets. 

Lusianthridin concentration dependently inhibited platelet aggregation stimulated by arachidonic acid, ADP, and collagen. These three agonists have their specific signaling pathways leading to platelet aggregation and they also share the common arachidonic acid-thromboxane pathway [16]. The ADP-induced platelet aggregation pathway includes two G protein-linked nucleotide receptors, namely P2Y1 and P2Y12. The P2Y1 receptor is linked to Gq, which leads to the activation of the β-isoform of phospholipase C (PLC). This receptor is responsible for changing platelet shape and mobilization of intracellular calcium, which causes initial aggregation (primary phase). P2Y12 is coupled to Gi for inhibition of adenylate cyclase to reduce cAMP levels. This receptor is responsible for the secretion of thromboxane A2 (TXA2) and other mediators, which act on their specific receptors and produce further platelet aggregation (secondary phase) [16]. These two phases of ADP-stimulated platelet aggregation can be detected by the turbidimetric aggregometry method [17]. P2Y1 receptor antagonists inhibit both primary and secondary phases of ADP-stimulated platelet aggregation and P2Y12 receptor antagonists inhibit the second phase without an effect on shape changes [18]. According to the current study, lusianthridin more selectively inhibited the second phase of ADP-stimulated platelet aggregation compared with the first phase. This result indicated that lusianthridin might not directly inhibit the pathway of P2Y1. In addition, the maximum percent inhibition of lusianthridin on ADP-induced platelet aggregation was only around 55%. Therefore, lusianthridin might not directly, but partially inhibit the pathway of P2Y12 (adenylate cyclase pathway). Moreover, lusianthridin increased cAMP levels in ADP-stimulated platelet aggregation. Prostaglandin E1 (PGE1) is a potent inhibitor of platelet aggregation. It directly stimulates adenylate cyclase, causing increased intracellular cAMP. The phosphodiesterase inhibitors inhibit the conversion of cAMP to AMP and increase the cAMP level, leading to inhibit calcium influx and secretion by suppressing 1,2-diacylglycerol (DAG), inositol 1,4,5 triphosphate (IP3), and TXA2 [19]. Therefore, the inhibitory effect of lusianthridin on ADP-induced suppression of cAMP formation might come from the inhibition of the adenylate cyclase pathway or inhibition of the phosphodiesterase enzyme. 

Collagen causes the activation of PLCγ2 producing IP3, which increases the release of calcium from the dense tubular system and DAG, which activates protein kinase C (PKC) [20]. The calcium release activates phospholipase A2 (PLA2), which causes a further increase of intracellular calcium level, secretion of granules and production of TXA2. This collagen-stimulated signaling pathway depends on TXA2 secretion to some extent [21]. Aspirin, a COX inhibitor, could inhibit collagen-stimulated platelet aggregation by inhibiting the TXA2 production [22]. In the present study, lusianthridin prolonged the lag time and inhibited collagen-stimulated platelet aggregation. Therefore, lusianthridin might act on calcium mobilization or TXA2 secretion to inhibit platelet aggregation. Further studies are needed in this area. 

Another agonist that we used in our study to induce platelet aggregation was arachidonic acid. Arachidonic acid is metabolized to prostaglandin (PG)H2 by the COX-1 enzyme in platelets. Then, PGH2 is converted to TXA2 by the peroxidase enzyme. TXA2 binds to two separate isoforms of TXA2 receptor, including TPα and TPβ that are linked to Gq and Gi, respectively [23]. This study showed that lusianthridin very potently inhibited platelet aggregation induced by arachidonic acid and prolonged the delay. Therefore, the antiplatelet activity of lusianthridin might be involved in the arachidonic acid metabolic pathway. 

Molecular docking and COX activity assays showed that lusianthridin probably inhibited both COX-1 and COX-2 enzymes by binding to the active sites of these enzymes. Lusianthridin seemed to bind the entrance site of the COX-1 enzyme (lobby region). Most conformations belonging to this site were characterized by the presence of a bond between Arg-120 and Glu-524. Amino acid residues including Pro-86, Ile-89, Leu-93, and Val-116 are assumed to be involved in the interaction of inhibitors with the lobby region [24]. In the present study, lusianthridin interacted with Tyr-355 by a pi-donor hydrogen bond and three amino acid residues (Ile-89, Leu-93, and Val-116) by hydrophobic bonds. Moreover, the key residues (Arg-120 and Glu-524) were surrounded by the binding site of lusianthridin. Therefore, lusianthridin probably inhibited the activity of COX-1 by blocking the lobby region of the enzyme, preventing its substrate, arachidonic acid, from reaching the catalytic site of the enzyme. For interaction with the COX-2 enzyme, lusianthridin did not interact closely with the binding site of arachidonic acid. However, lusianthridin formed hydrogen bonds with three amino acid residues (Pro-153, Arg-44, and Asn-39), which were probably the active region of the COX-2 enzyme interacted with isoindolines [25], caprolactam-salicylic ionic liquid [26], rofecoxib [27], naproxen modified derivatives [28], and aryl/heteroaryl substituted celecoxib derivatives [29]. This inhibitory effect of the COX-2 enzyme might be beneficial in platelet study, as a small amount of COX-2 is expressed inside the platelets [30]. In the previous findings, a COX-2 selective inhibitor reduced the production of thromboxane B2, although to a lesser amount compared to a COX-1 selective inhibitor [31].

This study revealed that the antiplatelet aggregation activities of lusianthridin probably act via arachidonic acid-thromboxane and adenylate cyclase pathways. Therefore, lusianthridin might be beneficial for the protection of atherosclerotic cardiovascular diseases. Lusianthridin also showed an inhibitory effect on the COX-2 enzyme. Further studies on its anti-inflammatory effects are warranted.

## 4. Materials and Methods

### 4.1. Materials and Chemicals

The chemicals: adenosine 5′-diphosphate sodium—ADP, trisodium citrate dihydrate, acetylsalicylic acid—ASA, dimethyl sulfoxide -DMSO, arachidonic acid, sigmacote, and other chemicals were purchased from Sigma-Aldrich Co., Saint Louis, MO, USA. Collagen type I was purchased from Chrono-Log, Havertown, PA, USA. Cyclooxygenase (COX) fluorescent inhibitor screening assay kit and cyclic adenosine monophosphate (cAMP) ELISA kit were purchased from Cayman Chemical, Ann Arbor, MI, USA. The tested compound, lusianthridin, was isolated from *Dendrobium venustum* and the purity was evaluated using NMR spectroscopy. Lusianthridin with more than 98% purity was used in this study [32].

### 4.2. Platelet-Rich Plasma (PRP) Preparation

This study was performed in human platelets from 18 healthy volunteers aged 18 to 50 years, with no prior or ongoing medical conditions. They were non-smokers, non-alcoholics, did not donate blood in the last one month, and did not take any medications at least 2 weeks before participating in the study. Volunteers signed the informed consent before participating in this study. This study was approved by the Institutional Review Board, Faculty of Medicine, Chulalongkorn University (COA No. 493/2020, approval date on 20 April 2020).

Blood samples (30 mL) from overnight fasting healthy volunteers were collected by venipuncture and put into plastic tubes containing 3.2% sodium citrate (blood: buffer = 9:1 *v*/*v*). The platelet-rich plasma (PRP) was prepared by centrifugation of blood samples at 21 °C, 200× *g* for 10 min and the top layer was collected as PRP. Isolation of the platelet-poor plasma (PPP) was done by further centrifugation of the rest of the blood samples at 21 °C, 1500× *g* for 15 min. PPP was used as a reference to define the theoretical point of 100% light transmission. Aggregation testing was done at least 15 min after PRP was prepared for recovering from refractoriness and used within 4 h of sample collection. 

### 4.3. Platelet Aggregation Measurement

Platelet aggregation test was performed as our previous study [33], which was modified from the Born turbidimetric aggregometry method [34] using an aggregometer (AggRAM™, Helena laboratories, Beaumont, TX, USA). PRP (200 µL) as 0% light transmission and PPP (250 µL) was set as 100% light transmission. Changes in light transmittance (at 600 nm wavelength) were recorded for 6 min after adding 25 µL of agonists. The submaximal concentration of agonists (the final concentration: ADP 4 µM, collagen 2 µg/mL, and arachidonic acid 0.5 µM) were used to induce platelet aggregation. PRP was pre-incubated with 25 µL of lusianthridin for 5 min at 37 °C (DMSO 0.5% as vehicle control and aspirin 0.1 mM as positive control). The concentrations of lusianthridin selected from the previous report of other phenanthrene derivatives [14] were used for initiation. The preliminary results showed that lusianthridin 0.1 mM have antiplatelet activities of 90, 58, and 87% inhibition when induced by arachidonic acid, ADP, and collagen, respectively. Subsequently, various concentrations of lusianthridin for each agonist were explored. Finally, the optimal concentration ranges were 0.0125 to 0.1 mM for arachidonic acid and 0.05 to 0.4 mM for ADP and collagen (data not shown). The study of each agonist was performed in five independent experiments. Each sample was done in triplicate. The change in light transmission was presented as the percentage of platelet aggregation.

### 4.4. Molecular Docking Study

AutoDock Vina (AutoDockTools 1.5.6., Scripps research, San Diego, CA, USA) [35], Discovery studio 2017 (Accerlrys, Inc., San Diego, CA, USA) and ChemBio 3D Ultra 12.0 (CambridgeSoft, Cambridge, MA, USA) were used to perform a Molecular docking study. From Protein Data Bank (PDB), the crystal structure of COX-1 (PDB ID: 2OYE) and COX-2 (PDB ID: 1CX2) were downloaded. The structure of lusianthridin was constructed by using ChemBioDraw. In the automated docking analysis, the active regions of COX-1 and COX-2 were defined in the grid box with 40 × 40 × 40 (1 Å spacing between grid points). The binding sites were determined using the interacted residues of the compound with COX enzymes and their respective binding energy. The lower binding affinity represents a higher inhibitory action. The docking results were visualized and analyzed using Discovery Studio software [33].

### 4.5. Cyclooxygenase Activity Assay

The tested compound, lusianthridin, was pre-treated directly with COX-1 or COX-2 enzymes for 5 min. Then, arachidonic acid and ADHP (10-acety-3,7-dihdroxyphenoxazine) were added for 2 min. Resorufin, a highly fluorescent compound, was produced from the reaction between prostaglandin G2 and ADHP. This fluorescence intensity could be analyzed with an excitation wavelength of 530–540 nm and an emission wavelength of 585–595 nm. The COX fluorescent inhibitor screening assay kit (Cayman) was used to perform this assay and the experiments were carried out according to the manufacturer’s instructions.

### 4.6. Measurement of cAMP Levels

PRP (4 × 108 platelets/mL) was pre-incubated with different concentrations of lusianthridin at 37 °C for 5 min with continuous stirring. A positive control, the 3-isobutyl-1-methylxanthine (IBMX), is a phosphodiesterase inhibitor used to prevent cAMP catabolism. After that, ADP (4 µmol/L) was added to induce platelet aggregation and the reaction was terminated by 0.5% ice-cold ethanol. The samples were then vortexed, sonicated, and centrifuged to obtain the supernatant [33]. The cAMP ELISA kit (Cayman) was used to perform this assay and the experiments were carried out according to the manufacturer’s instructions. The developed yellow color was analyzed at the wavelength of 405–450 nm.

### 4.7. Statistical Analysis 

All the experimental data were expressed as means ± standard errors of means (SEMs). Differences between groups were analyzed using one-way ANOVA followed by Dunnett’s post-hoc test with SPSS software version 22.0. *p*-values of less than 0.05 were considered to be statistically significant. Dose-response curves were analyzed by variable slope non-linear regression using GraphPad Prism 9.1 (GraphPad, San Diego, CA, USA) to obtain IC_50_ values.

## 5. Conclusions

This study revealed that lusianthridin, a phenanthrene derivative isolated from *Dendrobium venustum*, can inhibit platelet aggregation stimulated by arachidonic acid, collagen, and ADP. Its antiplatelet aggregation activities might act via arachidonic acid-thromboxane and adenylate cyclase pathways. Lusianthridin also showed inhibitory activity on COX-2 more potently than that on COX-1 enzyme, which might be beneficial for the anti-inflammatory effect. Further studies in animal models to explore the potentials in preventing atherothrombosis should be performed.

## Figures and Tables

**Figure 1 ijms-22-06846-f001:**
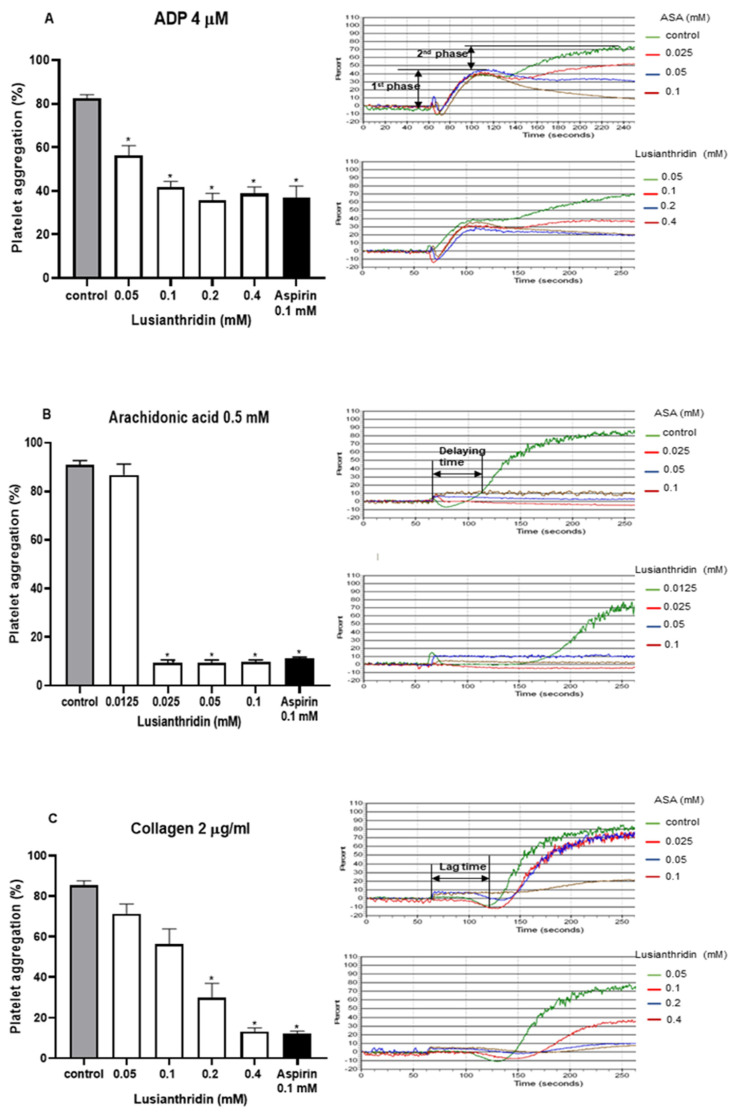
Effects of lusianthridin on agonist₋induced platelet aggregation. Platelets were pre₋incubated with 0.5% DMSO (vehicle control), lusianthridin, or aspirin at 37 °C for 5 min and then agonists were added to stimulate platelet aggregation. (**A**) ADP 4 µM. (**B**) Arachidonic acid 0.5 mM, the delaying time is the time starting from addition of an agonist until platelet aggregation. (**C**) Collagen 2 µg/mL, the lag time is a time starting from addition of an agonist until a platelet shape change. Data are presented as percent aggregation (means ± SEMs; *n* = 5). * *p* < 0.05 compared with those of vehicle control.

**Figure 2 ijms-22-06846-f002:**
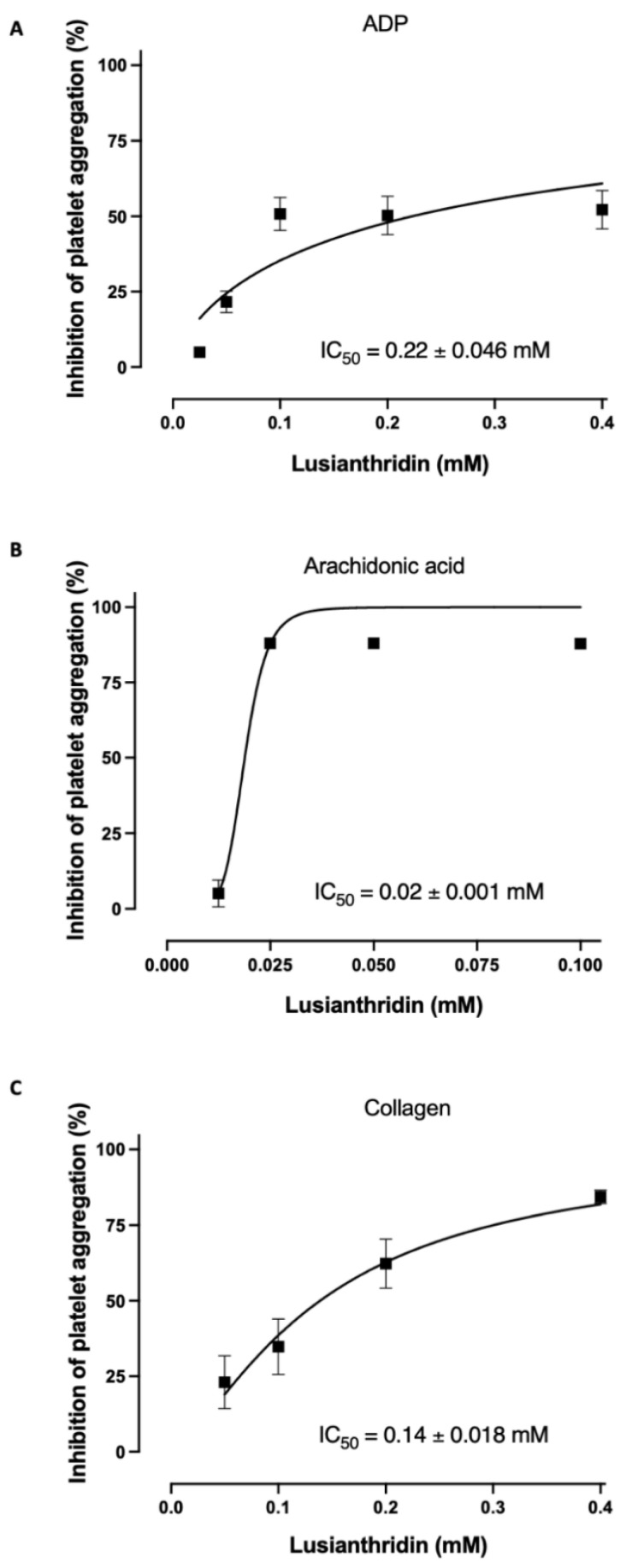
Lusianthridin concentration-dependently inhibited platelet aggregation induced by (**A**) ADP 4 µM, (**B**) Arachidonic acid 0.5 mM, and (**C**) Collagen 2 µg/mL. Data are presented as percent inhibition of platelet aggregation (means ± SEMs; *n* = 5).

**Figure 3 ijms-22-06846-f003:**
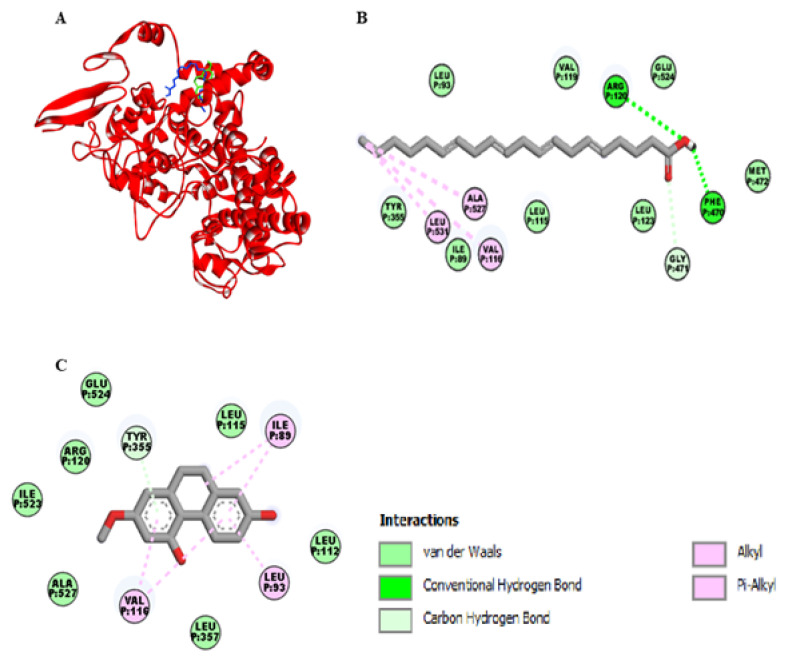
Docking lusianthridin into the COX-1 enzyme. (**A**) 3D interaction of arachidonic acid (blue) and lusianthridin (green) with the COX-1 enzyme. 2D interaction of binding mode for (**B**) arachidonic acid (**C**) lusianthridin inside the COX-1 enzyme.

**Figure 4 ijms-22-06846-f004:**
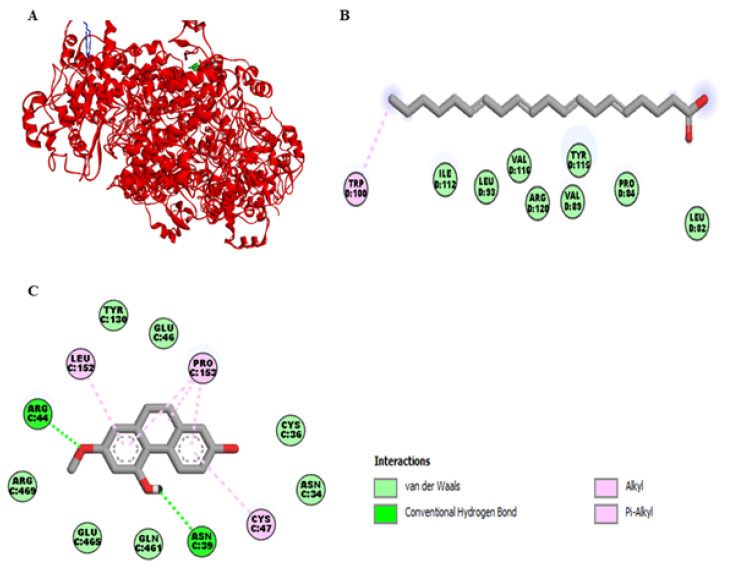
Docking lusianthridin into the COX-2 enzyme. (**A**) 3D interaction of arachidonic acid (blue) and lusianthridin (green) with the COX-2 enzyme. 2D interaction of binding modes for (**B**) arachidonic acid (**C**) lusianthridin inside the COX-2 enzyme.

**Figure 5 ijms-22-06846-f005:**
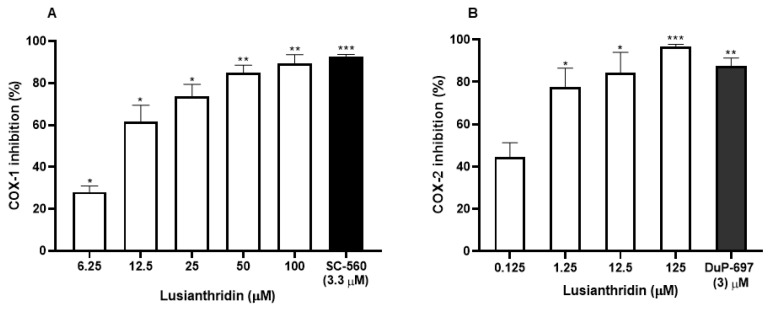
The inhibitory effects of various concentrations of lusianthridin on the activities of (**A**) COX-1 and (**B**) COX-2 enzymes. Tested compounds were incubated directly with COX-1 or COX-2 in the assay buffer for 5 min and then the activities were measured by a COX fluorescent inhibitor screening assay kit. SC-560, a COX-1 selective inhibitor, and DuP-697, a COX-2 selective inhibitor, were used as positive controls. Data are presented as means ± SEMs (*n* = 3). * *p* < 0.05, ** *p* < 0.001, *** *p* < 0.0001 compared with control.

**Figure 6 ijms-22-06846-f006:**
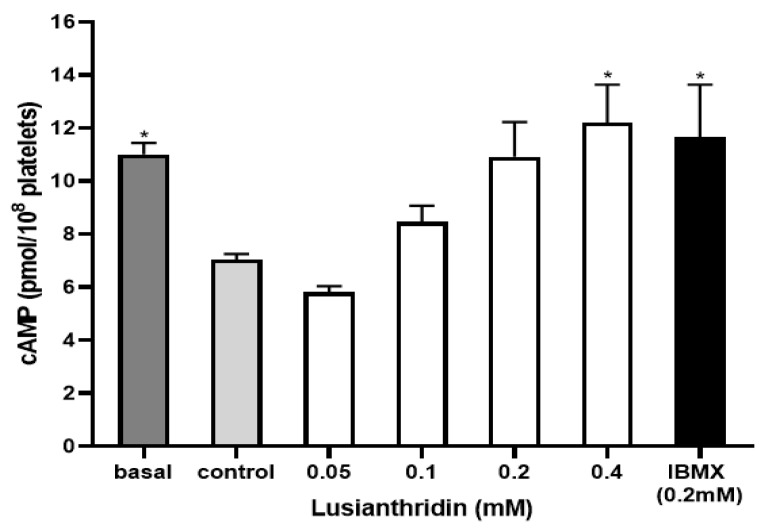
The effects of lusianthridin on cAMP levels in platelets. Platelets were pre-incubated with various concentrations of compounds for 5 min with continuous stirring at 37 °C and ADP (4 µM) was then added to induce platelet aggregation. An ELISA kit was used to measure cAMP levels. IBMX, a phosphodiesterase inhibitor, was used as a positive control. Data are presented as means ± SEMs (*n* = 4). * *p* < 0.05 compared with control.

## Data Availability

The datasets used and analyzed during the current study are available from the corresponding author on reasonable request.

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
