# Peer review of "Inhibitory Mechanisms of Lusianthridin on Human Platelet Aggregation"

_ijms, 2021, doi:10.3390/ijms22136846_

Round 1

Reviewer 1 Report

Dendrobium represents one of the most important orchid genera in the medicine. The study of anti-platelet action of the compounds found in the Dendrobium have been carried out long time ago (i.e. 10.1055/s-2008-1074492) and is nothing new in the scientific word. The authors of presented manuscript have published already several papers describing the activities of phenolic compounds from Dendrobium which make them experts in the field but also put on them higher requirements towards determination and enhancement of the novelty of the research. Presenting the revised and updated manuscript that is suitable for publication in International Journal of Molecular Sciences will require also attaching the comments below:
1. How the concentration of the Lusianthridin were chosen? There are several tested concentration and little is shown about their toxicity (the highest concentration does not equal the best one)

  1. Why the comparison with the clopidogrel was not performed? Only the data comparing the results for Lusianthridin and aspirin were presented.
  2. How many samples (human samples) were there included in each study and control groups?
  3. The conclusions are not sufficient to discuss all the findings presented in the paper.

After the improvement of the manuscript it could be further concider to be published in IJMS.

Reviewer 2 Report

Please see attached comments.

Round 2

Reviewer 1 Report

Thank you for the reply. All of my comments have been adressed.

Reviewer 2 Report

The authors addressed the concerns in the review and the work is acceptable for publication.